# TEMPORAL REPETITION COUNTING WITH DYNAMIC ACTION QUERIES

## ABSTRACT

Temporal repetition counting aims to quantify the repeated action cycles within a video. The majority of existing methods rely on the similarity correlation matrix to characterize the repetitiveness of actions, but their scalability is hindered due to the quadratic computational complexity. In this work, we introduce a novel approach that employs an action query representation to localize repeated action cycles with linear complexity. Based on this representation, we further develop two key components to tackle the fundamental challenges of temporal repetition counting. Firstly, to facilitate open-set action counting, we propose the *dynamic action query*. Unlike static action queries, this approach dynamically embeds video features into action queries, offering a more flexible and generalizable representation. Second, to distinguish between actions of interest and background noise actions, we incorporate *inter-query contrastive learning* to regularize the video feature representation corresponding to different action queries. The experiments demonstrate that our method significantly outperforms the state-of-the-art methods in terms of both accuracy and efficiency. Specifically, our approach exhibits versatility in handling long video sequences, unseen actions, and actions at various speeds across two challenging benchmarks. [1]

## 1 INTRODUCTION

Temporal periodicity is a ubiquitous phenomenon in the natural world. Temporal Repetition Counting (TRC) aims to accurately measure the number of repetitive action cycles within a given video and holds significant potential for applications ranging from pedestrian detection (Ran et al., 2007) to fitness monitoring (Fieraru et al., 2021).

Pioneer methods (Laptev et al., 2005; Azy & Ahuja, 2008; Cutler & Davis, 2000; Tsai et al., 1994; Pogalin et al., 2008; Thangali & Sclaroff, 2005; Chetverikov & Fazekas, 2006) represent time-series video data as one-dimensional signals and employ spectral analysis techniques such as the Fourier transform. While suitable for short videos with fixed periodic cycle lengths, these methods struggle to handle real-world scenarios with varying cycle lengths and sudden interruptions. Recent studies shift to deep learning-based methods (Levy & Wolf, 2015; Dwibedi et al., 2020; Hu et al., 2022; Zhang et al., 2020) and show promising performance. Notably, most of these methods, such as RepNet (Dwibedi et al., 2020) and TransRAC (Hu et al., 2022), utilize a temporal similarity correlation matrix to depict repetitiveness, as illustrated in Figure 1 (a). Nevertheless, the computational complexity of this representation grows quadratically with the number of input frames, limiting the model scalability. As a result, these methods struggle to simultaneously accommodate varying action periods and video lengths.

Recent progress in the field of Temporal Action Detection (TAD) (Zhang et al., 2022a; Liu et al., 2022b) introduces an efficient representation of action periods by associating each action instance with an action query, similar to DETR (Carion et al., 2020). Inspired by this, we propose to formulate the TRC problem as a *set prediction* task and the goal is to detect every action cycle by representing it as an action query. Figure 1 (b) shows an overview of our method. This novel formulation reduces the complexity from quadratic to linear and enables counting long videos with varying action periods. However, directly applying DETR-like TAD approaches to the TRC problem proves inadequate in

---

[1]Code and models will be publicly released.

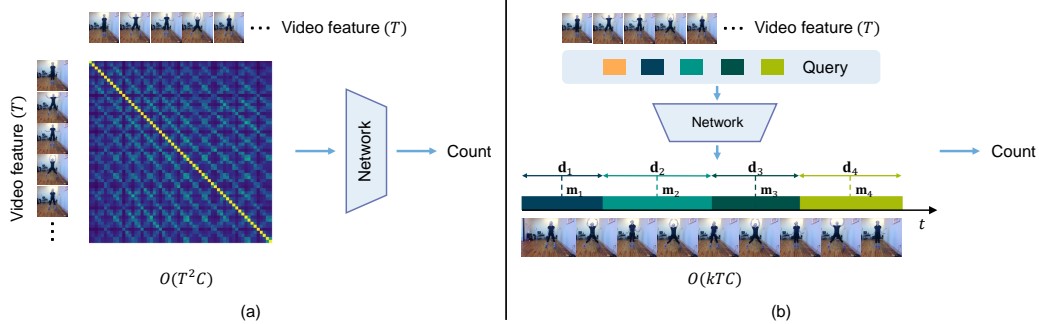

Figure 1: **Comparison of (a) similarity matrix-based approaches and (b) our action query-based approach.** Unlike similarity matrix-based approaches, which have a quadratic increase in computation complexity with input video length, our approach achieves a linear increase.

addressing two distinctive challenges unique to TRC task, resulting in inferior performance as shown in Section 4.3 and 4.4. We highlight the two inherent differences between TRC and classic detection problems (including TAD):

1. TRC requires recognizing *open-set* action instances depending on the input video, rather than relying on predefined class labels.

2. TRC requires recognizing action instances with *identical* content, while detection does not.

In response to the first challenge, we propose the *Dynamic Action Query* (DAQ), which adaptively updates the action query using distilled content features from the encoder. This mechanism allows the decoder to attend to the action of interest based on the input video contents. To tackle the second challenge, we propose an *Inter-query Contrastive Learning* (ICL) strategy. It enforces that repetitive action cycles exhibit similar representations, while simultaneously having dissimilar representations with non-repetitive (background) video content. Extensive experiments validate the effectiveness of our proposed designs.

We summarize our contributions as follows:

- We provide a novel perspective to tackle the TRC problem using a simple yet effective representation for action cycles. This approach reduces the computational complexity from quadratic to linear and proves robust to varying action periods and video lengths.

- We propose *Dynamic Action Query* to guide the model to focus on the action of interest and improve generalization ability across different actions.

- We introduce *Inter-query Contrastive Learning* to facilitate learning repetitive action representations and to distinguish these from distractions.

- Our experiments on two challenging benchmarks demonstrate the superiority of our method, showing a significant improvement compared to state-of-the-art methods. Notably, our method strikes an effective balance between handling various action periods and video lengths.

## 2 RELATED WORK

### 2.1 TEMPORAL REPETITION COUNTING

Traditional methods (Laptev et al., 2005; Azy & Ahuja, 2008; Cutler & Davis, 2000; Tsai et al., 1994; Pogalin et al., 2008; Thangali & Sclaroff, 2005; Chetverikov & Fazekas, 2006) frequently employ spectral or frequency domain techniques for the analysis of repetitive sequences, thereby preserving the underlying repetitive motion structures. While these conventional approaches are capable of effectively handling simple motion sequences or those characterized by fixed periodicity, they prove inadequate when confronted with non-stationary motion sequences encountered in real-world scenarios. In contrast, deep-learning-based approaches (Levy & Wolf, 2015; Dwibedi et al.,

2020; Hu et al., 2022; Zhang et al., 2020) have demonstrated remarkable performance improvements. Notably, RepNet (Dwibedi et al., 2020) and TransRAC (Hu et al., 2022) leverage temporal similarity matrices of actions to construct models for counting temporal repetitions. However, these similarity-matrix-based methods are not scalable for long videos due to their quadratic computational complexity. Another research line involves predicting the start and end points of each cycle (Zhang et al., 2020) from coarse to fine. Nevertheless, its practicality is hindered by the requirement for over 30 forward passes to count iteratively from a single video. In this paper, we introduce an effective action cycle representation by leveraging a Transformer encoder-decoder, which reduces the computational complexity from quadratic to linear and demonstrates superior performance in handling both fast and slow actions.

## 2.2 TEMPORAL ACTION DETECTION

The field of temporal action detection (Redmon et al., 2016; Zhao et al., 2017; Chao et al., 2018) is typically classified into two categories: anchor-based methods, and anchor-free methods. Anchor-based methods (Zeng et al., 2019; Li & Yao, 2021; Qing et al., 2021) generate multiple anchors, subsequently classifying these anchors to determine the action boundaries. Anchor-free methods (Buch et al., 2019; Shou et al., 2017; Yuan et al., 2017; Lin et al., 2021) predict action instances by directly regressing the boundary and the center point of an action instance. With the rapid development of Transformer technology, DETR (Carion et al., 2020) is introduced for object detection task (Zhu et al., 2020; Meng et al., 2021; Liu et al., 2022a; Zhang et al., 2022b) and gains increasing popularity with promising performance. This paradigm promotes the study in many fields such as the action detection tasks (Liu et al., 2022b; Tan et al., 2021; Vaswani et al., 2017; Wang et al., 2021). These methods establish a direct connection between action queries and the predicted action instances, enabling them to accurately predict the temporal boundaries of actions. Inspired by these promising results, we explore the possibility of utilizing a novel action query to represent the action cycle in TRC task. In contrast to existing TAD methods, our approach allows the model to capture the inherent repetitive content of an action cycle without relying on predefined class labels and effectively addresses confounding factors such as non-repetitive video backgrounds. This makes our approach well-suited for tackling the challenges of the TRC problem.

## 3 METHOD

### 3.1 OVERVIEW

Given an RGB video sequence $\mathbf{V}$ with $T$ frames, our model predicts an integer $N$ indicating the number of estimated repetitive action cycles. The entire model consists of the backbone network $\Phi(\cdot)$, the Transformer network with encoder $\mathcal{E}(\cdot)$, decoder $\mathcal{D}(\cdot)$, and multiple prediction heads as shown in Figure 2. The backbone network $\Phi(\cdot)$ takes a sequence of $T$ video frames as input and extracts feature vectors $\mathbf{F} \in \mathbb{R}^{T \times C}$ for each frame using widely-used backbone networks such as TSN (Wang et al., 2016), where $C$ denotes the feature dimension.

We leverage a Transformer encoder-decoder architecture to process the features $\mathbf{F}$ and detect each action cycle instance to obtain the action count. The core idea is to connect the action instances with a set of *action queries*, constructed by *content queries* $\mathbf{Q}^{\mathrm{cnt}} \in \mathbb{R}^{Q \times C}$ and *position queries* $\mathbf{Q}^{\mathrm{pos}} \in \mathbb{R}^{Q \times C}$, where $Q$ denotes the number of queries. The content queries are responsible for the action category, while the position queries handle the action time period. In order to automatically detect the open-set action categories of interest, we propose the *Dynamic Action Query (DAQ)* strategy, which adaptively updates the action query of the decoder using distilled content features from the encoder. We leave the position queries as learnable parameters. Through iterative refinement by the decoder and two prediction heads, these queries are updated to accurately align with the corresponding time periods of repetitive actions. The two prediction heads include a classification head and a position head. The classification head is to assign a class label for each content query, with $K = 2$ classes consisting of actions and null ($\emptyset$). The position head network transforms the decoder output features $\mathbf{H}^{\mathrm{dec}}$ to a set of time positions bias $\Delta \mathbf{P}$, which are used to iteratively get final time periods $\mathbf{P}$, including the midpoints and duration of each action cycle. To recognize action instances with identical content, we propose *Inter-query Contrastive Learning (ICL)* to guide the network to focus on repetitive actions while getting rid of other distractors. This strategy clusters the queries into the positive action set ($S^+$) and negative null set ($S^-$) according to the predicted class

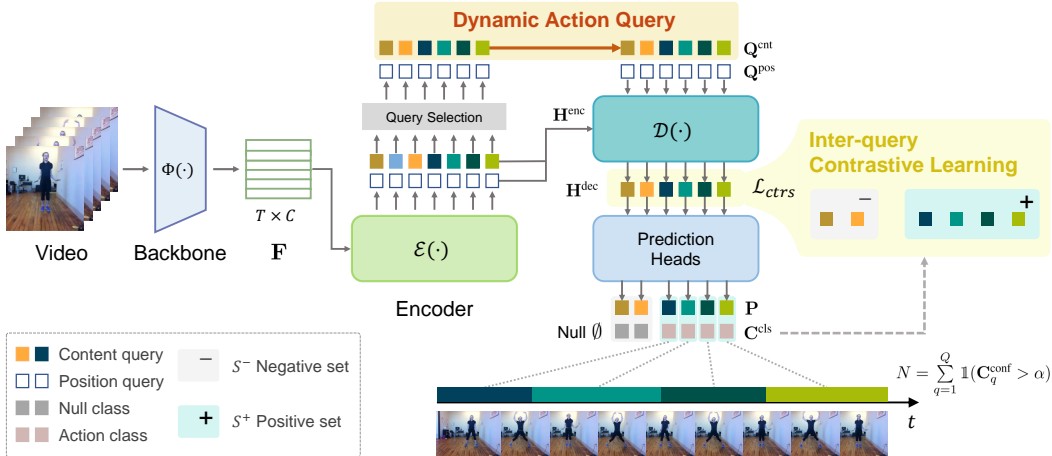

Figure 2: **Framework overview.** Given the video input, we first extract video features with the backbone network and then feed them to the encoder. The encoder processes the features $\mathbf{F}$ and selects top-$Q$ queries to be the dynamic content queries $\mathbf{Q}^{cnt}$ for the decoder. The decoder refines the queries along with learnable position queries $\mathbf{Q}^{pos}$ and maps them to the action instances with the prediction heads and predicts the final action count. We apply inter-query contrastive learning on the decoder features $\mathbf{H}^{dec}$.

label $\mathbf{C}^{cls}$. Finally, we obtain the count value $N$ by selecting the action instances whose confidence scores $\mathbf{C}^{conf}$ are above a certain threshold $\alpha$ according to the classification results. Both the encoder and decoder are Transformer networks with $\mathcal{L}^{enc}$ and $\mathcal{L}^{dec}$ layers.

## 3.2 MODEL ARCHITECTURE

**Encoder and query selection.** The encoder $\mathcal{E}(\cdot)$ refines the video features $\mathbf{F}$ extracted by the backbone network and feeds them to a query selection module. The query selection module selects $Q$ queries from the output features of $\mathcal{E}(\cdot)$ to be the initial content queries $\mathbf{Q}^{cnt}$ for the decoder based on their confidence scores. The confidence scores are obtained by applying the classification head network to these features.

**Decoder.** The decoder $\mathcal{D}(\cdot)$ employs a layer-by-layer update mechanism to output the final position of the action instance of interest. We use the midpoint and the period duration of each action cycle to denote each action instance. The initial action position $\mathbf{P}^{init} = (\mathbf{m}^{init}, \mathbf{d}^{init}) \in \mathbb{R}^{Q \times 2}$ is obtained from the learnable position queries $\mathbf{Q}^{pos}$ with a linear layer. The decoder takes the output features $\mathbf{H}^{enc}$ of the encoder, the content query $\mathbf{Q}^{cnt}$, and the initial action position $\mathbf{P}^{init}$ as input and outputs $\mathbf{H}^{dec} \in \mathbb{R}^{Q \times C}$ as well as update the action position layer-by-layer. We apply a position head and a classification head network to process the output feature $\mathbf{H}^{dec}$, respectively, and finally get the action instances, which are formed with the refined position $\mathbf{P} \in \mathbb{R}^{Q \times 2}$ and the class label $\mathbf{C}^{cls} \in \mathbb{R}^{Q \times K}$.

**Prediction heads.** The prediction heads consist of a classification head network and a position head network. The classification head is a fully-connected network. Given that the action category is an open set and not preset, we set the number of classes $K$ to be 2 in the classification head, indicating whether the query is the action instance of interest or a null ($\emptyset$) class. The classification head produces the probability $\mathbf{C}^{cls}$ for each action instance with the Softmax function. The maximum probability value serves as their confidence score $\mathbf{C}^{conf}$. The position head, which is a two-layer MLP, computes the residual changes of the midpoint and duration of the action positions using the intermediate features $\mathbf{H}^{dec}$ generated by each decoder layer. The action positions for the next decoder layer are obtained by adding the predicted residual to the action positions of the previous layer. The query selection module reuses the class head and position head, and these two networks share parameters with the decoder. We apply binary Cross-entropy loss to supervise the predicted class labels $\mathbf{C}^{cls}$ for each action instance, aiming to guide the model in effectively discriminating the action instances of

interest from others:

$$\mathcal{L}_{cls} = \sum_{q=1}^{Q} \text{CrossEntropyLoss}\left(\mathbf{C}_q^{\text{cls}}, \tilde{\mathbf{C}}_q^{\text{cls}}\right) = \sum_{q=1}^{Q} \sum_{i=1}^{K} -\tilde{c}_{q,i} \log\left(c_{q,i}\right), \tag{1}$$

where $\mathbf{C}_q^{\text{cls}} = [c_{q,1}, c_{q,2}, ..., c_{q,K}] \in \mathbb{R}^K$ represents the predicted class label for $q^{th}$ action query. ˜ denotes the GT label.

We also supervise the predicted action positions with L1 loss and gIoU (Rezatofighi et al., 2019) loss:

$$\mathcal{L}_{pos} = \sum_{q=1}^{Q} \left\| \mathbf{P}_q - \tilde{\mathbf{P}}_q \right\| + \lambda_{gIoU} \sum_{q=1}^{Q} \left(1 - \text{gIoU}(\mathbf{P}_q, \tilde{\mathbf{P}}_q)\right), \tag{2}$$

where $\mathbf{P}_q = (\mathbf{m}_q, \mathbf{d}_q)$ represents the predicted action instance position for the $q^{th}$ query, which consists of the midpoint and duration of the action instance. ˜ stands for GT label.

### 3.3 DYNAMIC ACTION QUERY

As discussed in Section 1, the TRC problem requires to recognize *open-set* action instances depending on the video content, while the action category is not predefined. Therefore, we decompose the query into the content query and position query and propose the *Dynamic Action Query*, which adaptively updates the content (action) query using distilled content features from the encoder, as shown in Figure 2.

Concretely, we first perform query selection on the encoder output features $\mathbf{H}^{\text{enc}} \in \mathbb{R}^{T \times C}$ and select top-$Q$ query features as priors for the decoder which contains the video content features. We re-use the two prediction heads to process $\mathbf{H}^{\text{enc}}$ to extract the action class and the action position. Then we select the top-$Q$ features with high confidence over all the action queries, excluding the null query according to the classification results. These selected queries finally serve as the initial content query $\mathbf{Q}^{\text{cnt}}$ for the decoder.

In this way, the content query is enhanced by the extracted content-related features from the encoder which contains the action priors of interest. It helps the decoder to focus more on the substantial action features from the encoder, thereby achieving stronger generalization capability. We also explore several other updating strategies for different types of queries to validate the effectiveness of DAQ. Please refer to Section 7.1 in the supplementary for details.

### 3.4 INTER-QUERY CONTRASTIVE LEARNING

Another unique challenge to the TRC task is to recognize action instances with *identical* content. To tackle this challenge, we propose *Inter-query Contrastive Learning*. Intuitively, queries corresponding to action cycles should have similar representations, while other queries should have dissimilar representations.

Specifically, the classification head takes the decoder features $\mathbf{H}^{\text{dec}}$ of the $Q$ queries as input and classifies them into $K = 2$ classes, consisting of *actions* and *null* $\emptyset$. Then we use the Hungarian matching algorithm (Carion et al., 2020) to match the $Q$ disordered predicted instances with the GT labels. We define the matched action instances as a set of positive samples $S^+$, while the rest null instances as a set of negative samples $S^-$. We apply contrastive learning over the $Q$ decoder features $\mathbf{H}^{\text{dec}}$ using InfoNCE loss (He et al., 2020):

$$\mathcal{L}_{ctrs} = -\sum_{q \in S^+} \log\left(\frac{\sum_{s \in S^+, s \neq q} \exp(\mathbf{H}_q^{\text{dec}} \cdot \mathbf{H}_s^{\text{dec}})/\tau}{\sum_{s \in S^+, s \neq q} \exp(\mathbf{H}_q^{\text{dec}} \cdot \mathbf{H}_s^{\text{dec}})/\tau + \sum_{s \in S^-} \exp(\mathbf{H}_q^{\text{dec}} \cdot \mathbf{H}_s^{\text{dec}})/\tau}\right), \tag{3}$$

where $\tau$ is the temperature parameter, and $\cdot$ denotes inner product.

### 3.5 TRAINING LOSS

We train the whole network in a supervised way. The overall loss function is defined as:

$$\mathcal{L} = \lambda_{cls}\mathcal{L}_{cls} + \lambda_{pos}\mathcal{L}_{pos} + \lambda_{ctrs}\mathcal{L}_{ctrs}, \tag{4}$$

where $\lambda_{cls}, \lambda_{pos}, \lambda_{ctrs}$ are the coefficients of each loss term, respectively.

## 4 EXPERIMENTS

### 4.1 DATASETS AND METRICS

**RepCountA dataset** (Hu et al., 2022) is primarily gathered from fitness videos uploaded on YouTube. It features longer video lengths, greater variations in average motion cycle changes, and more repetitive cycles than previous evaluation datasets (Dwibedi et al., 2020; Runia et al., 2018; Levy & Wolf, 2015; Zhang et al., 2020). The dataset contains fitness activities performed in various environments such as home, gym, and outdoors. The activities include push-ups, pull-ups, jumping jacks, *etc*. The dataset contains 1041 videos, with 757 videos allocated to the training set, 130 to the validation set, and 151 to the test set. The maximum video length and counting number in this dataset are 88 seconds and 141, respectively. We use the start and end position of each action instance provided from annotations as $\hat{\mathbf{P}}_q$, and set the class label $\hat{\mathbf{C}}_q^{cls}$ of all instance to be 1. We train our model on the RepCountA train set and select the best-performed model on the validation set. We report the evaluation results on the test set.

**UCFRep dataset** (Zhang et al., 2020) is a subset of the UCF101 dataset (Soomro et al., 2012), including fitness videos and daily life videos. The dataset includes 526 videos, with 420 videos in the training set and 106 videos in the validation set. The maximum video length and counting number in this dataset are approximately 34 seconds and 54, respectively. Following previous work (Hu et al., 2022), we do not use the train set but directly test our model on the test set to evaluate the model generalization ability.

**Metrics.** We compute two commonly used metrics to evaluate the model performance, including OBO and MAE (Hu et al., 2022; Dwibedi et al., 2020; Zhang et al., 2020). **OBO** (Off-By-One count error) measures the probability that the predicted count is within 1 unit of the GT count, and considers a prediction to be correct if it falls within this range. **MAE** (Mean Absolute Error) measures the normalized absolute difference between the predicted and GT counts. Formally,

$$\text{OBO} = \frac{1}{M} \sum_{i=1}^{M} \left| N_i - \hat{N}_i \leq 1 \right|, \quad \text{MAE} = \frac{1}{M} \sum_{i=1}^{M} \frac{\left| N_i - \hat{N}_i \right|}{N_i}, \tag{5}$$

where $N_i$ and $\hat{N}_i$ are the predicted and GT counts for the $i^{th}$ test video, respectively, and $M$ is the total number of test videos.

To better evaluate the performance of different models in recognizing actions with varying periods, we expand the evaluation metrics with three variants for the OBO and MAE metrics. We split the test video set into three categories based on the average single action period time: short-, medium-, and long-period test sets. We define videos with an average single action duration of fewer than 30 frames as belonging to the short-period test set, videos with an average action duration longer than 60 frames as the long-period test set, and the remaining videos as belonging to the medium-period test set. We report the OBO and MAE metrics on each of these sets separately.

### 4.2 IMPLEMENTATION DETAILS

We employ the backbone network with commonly-used TSN (Wang et al., 2016) RGB branch or VideoMAE (Tong et al., 2022) networks, which are both pre-trained on the Kinetics400 (Kay et al., 2017) dataset. For the Transformer architecture, we employ a 2-layer encoder and a 4-layer decoder, both with 8-head attention mechanisms. The feature dimension is set to $C = 512$. We use $Q = 40$ queries in the model. The length of the video input is set to $T = 512$ frames without down-sampling. We utilize the AdamW optimizer (Loshchilov & Hutter, 2017) with a learning rate of 0.002, a batch size of 64, and train the model for 80 epochs. $\lambda_{cls}$, $\lambda_{pos}$, $\lambda_{gIoU}$, and $\lambda_{ctrs}$ are set to be 1.0, 5.0, 0.4, 1.0, respectively. The action instance confidence threshold $\alpha$ is 0.2.

### 4.3 COMPARISON TO STATE-OF-THE-ARTS

**Results on RepCountA dataset.** We compare our proposed approach to the state-of-the-art methods on the RepCountA (Hu et al., 2022) dataset, and the results are presented in Table 1. Our approach achieves the best performance over the whole test set, outperforming previous works by 70.8% on

Table 1: **Comparison to the state-of-the-arts on RepCountA (Hu et al., 2022) dataset.** † denotes the result by sequentially segmenting the video into the needed input length for each method. ActionFormer (Zhang et al., 2022a) and TadTR (Liu et al., 2022b) are trained by treating each action category as a distinct class, while ActionFormer* and TadTR* represent models trained by treating all action categories as a single class. We additionally report MAE and OBO metrics for short-, medium-, and long-period test sets.

| | MAE ↓ | OBO ↑ | $MAE_s$ ↓ | $OBO_s$ ↑ | $MAE_m$ ↓ | $OBO_m$ ↑ | $MAE_l$ ↓ | $OBO_l$ ↑ |
|---|---|---|---|---|---|---|---|---|
| RepNet (Dwibedi et al., 2020) | 0.5865 | 0.2450 | 0.7793 | 0.0930 | 0.5893 | 0.1591 | 0.4549 | 0.4062 |
| TransRAC (Hu et al., 2022) | 0.4891 | 0.2781 | 0.5789 | 0.0233 | 0.4696 | 0.2955 | 0.4420 | 0.4375 |
| ActionFormer (Zhang et al., 2022a) | 0.4833 | 0.2848 | 0.4106 | 0.0930 | 0.3816 | 0.2955 | 0.6021 | 0.4062 |
| ActionFormer* (Zhang et al., 2022a) | 0.4990 | 0.2781 | 0.4164 | 0.1628 | 0.3768 | 0.3409 | 0.6385 | 0.3125 |
| TadTR (Liu et al., 2022b) | 0.9306 | 0.2053 | 0.8814 | 0.0233 | 0.7230 | 0.1591 | 1.1063 | 0.3594 |
| TadTR* (Liu et al., 2022b) | 1.1314 | 0.0662 | 0.8364 | 0.0233 | 1.1591 | 0.0000 | 1.3106 | 0.1406 |
| RepNet (Dwibedi et al., 2020) † | 0.6734 | 0.3311 | **0.1222** | **0.4419** | 0.2926 | 0.2727 | 1.3056 | 0.2969 |
| Context (Zhang et al., 2020) † | 0.5257 | 0.3179 | 0.3185 | 0.2093 | 0.3626 | 0.3182 | 0.7771 | 0.3906 |
| TransRAC (Hu et al., 2022) † | 2.5842 | 0.0728 | 1.0150 | 0.2093 | 2.0664 | 0.0000 | 3.9945 | 0.0312 |
| Ours (TSN) | 0.2809 | 0.4570 | 0.2411 | 0.1628 | **0.1792** | 0.5455 | 0.3776 | 0.5938 |
| Ours (VideoMAE) | **0.2622** | **0.5430** | 0.2257 | 0.2558 | 0.2002 | **0.5909** | **0.3294** | **0.7031** |

Table 2: **Generalization comparison on UCFRep (Zhang et al., 2020) dataset.** We additionally report MAE and OBO metrics for short-, medium-, and long-period test sets.

| | MAE ↓ | OBO ↑ | $MAE_s$ ↓ | $OBO_s$ ↑ | $MAE_m$ ↓ | $OBO_m$ ↑ | $MAE_l$ ↓ | $OBO_l$ ↑ |
|---|---|---|---|---|---|---|---|---|
| RepNet (Dwibedi et al., 2020) | **0.5336** | 0.2984 | 0.6219 | 0.1739 | 0.4825 | 0.3600 | 0.4996 | 0.5000 |
| TransRAC (Hu et al., 2022) | 0.6180 | 0.3143 | 0.6296 | **0.1951** | 0.5842 | 0.4250 | 0.6784 | 0.4118 |
| Ours (TSN) | 0.6016 | 0.2959 | 0.7069 | 0.0488 | 0.5777 | 0.4250 | **0.4039** | 0.5882 |
| Ours (VideoMAE) | 0.5435 | **0.4184** | **0.5657** | **0.1951** | **0.4625** | **0.5500** | 0.6804 | **0.6471** |

the OBO metric. In particular, our approach gains high performance on the medium- and long-period actions, outperforming competing methods significantly. Previous works struggle to achieve a balance and show limited effectiveness when dealing with actions of different lengths. Especially, the TransRAC (Hu et al., 2022) yields an inferior performance on short-period video for the metric $OBO_s$ due to its sparse sampling of the entire video, resulting in the loss of multiple cycles.

For a fair comparison, we re-evaluate the previous methods using a sliding-window strategy, in which the original untrimmed video is divided into segments, and the final counting result is obtained by summing the counting result of each segment. The results are shown with † in Table 1. It can be observed that even though RepNet (Dwibedi et al., 2020) achieves accurate counting for short actions, it still faces challenges in recognizing long actions. The performance on the short-period actions may be attributed to the local feature matching proposed in RepNet (Dwibedi et al., 2020). On the other hand, the results obtained by TransRAC (Hu et al., 2022) are even less satisfactory.

We additionally compare our method with two state-of-the-art TAD methods, ActionFormer (Zhang et al., 2022a) and TadTR (Liu et al., 2022b). To adapt the TAD methods for TRC task, we explore two training strategies by treating each action category as a distinct class or a single class (denoted with * in Table 1). The inferior performance of these TAD methods is attributed to the distinct difference between the TAD and TRC tasks and therefore demonstrates the effectiveness of our approach.

However, our method could be improved in the detection of short-period actions in future work, which is similar to the challenge of recognizing small objects in the field of object detection (Carion et al., 2020).

**Results on UCFRep dataset.** We evaluate the generalization ability of our method in Table 2, where we directly use the model trained on the RepCountA dataset (Hu et al., 2022) and evaluate it on the validation set of the UCFRep dataset(Zhang et al., 2020) following previous work (Hu et al., 2022). Our approach gets consistent improvement compared to existing works. The improvement becomes more significant on the longer-period actions, validating the effectiveness and the strong generalization ability of our method.

Table 3: **Effect of different modules**. DAQ means the use of dynamic action query, and ICL means inter-query contrastive learning by using contrastive loss.

| | MAE $\downarrow$ | OBO $\uparrow$ | MAE$_s$ $\downarrow$ | OBO$_s$ $\uparrow$ | MAE$_m$ $\downarrow$ | OBO$_m$ $\uparrow$ | MAE$_l$ $\downarrow$ | OBO$_l$ $\uparrow$ |
|---|---|---|---|---|---|---|---|---|
| Baseline | 0.4592 | 0.3509 | 0.2805 | 0.1576 | 0.3037 | 0.4318 | 0.6862 | 0.3906 |
| (a) + DAQ | 0.4035 | 0.4040 | 0.2448 | **0.2093** | 0.3106 | 0.4545 | 0.5740 | 0.5000 |
| (b) + ICL | 0.3542 | 0.4172 | 0.2624 | **0.2093** | 0.2515 | 0.5227 | 0.4864 | 0.4844 |
| (c) + DAQ + ICL | **0.2809** | **0.4570** | **0.2411** | 0.1628 | **0.1792** | **0.5455** | **0.3776** | **0.5938** |

Table 4: **Comparison of model complexity.** Due to variations in input video lengths and the frozen backbones used in both methods, We compare the computational complexity of the backbone when processing a single frame and the computational complexity of the rest counting part when processing the input sequences.

| | Input length | Backbone (per frame) | | Counting part (all frames) | |
|---|---|---|---|---|---|
| | | Params (M) | FLOPs (G) | Params (M) | FLOPs (G) |
| RepNet (Dwibedi et al., 2020) | 64 | 23.51 | 1.11 | 20.47 | 92.40 |
| TransRAC (Hu et al., 2022) | 64 | 27.85 | 7.53 | 14.43 | 100.19 |
| Ours (TSN) | 512 | 23.51 | 5.38 | 16.81 | 3.55 |

## 4.4 ABLATION STUDY

**Effect of different modules.** Table 3 shows the results on the RepCountA (Hu et al., 2022) dataset to ablate the effect of each proposed module. The baseline method uses a static query and does not incorporate inter-query contrastive learning by not using the contrastive loss. The static query means removing the DAQ module where both the content query and the position query are learnable variables instead of updating from the encoder output. As a result of using the query-based representation, the baseline model is able to naturally handle long-time videos, leading to comparable results with the state-of-the-art method TransRAC (Hu et al., 2022). When (a) changing to the proposed DAQ in the baseline, we obtain consistent improvement to the static query and get larger improvement on the OBO$_l$ metric. When (b) adding the ICL strategy to the baseline, we observe similar improvements. These two modules show similar improvements in detecting short-period actions. ICL performs better in recognizing actions with middle-period lengths, while DAQ exhibits a noticeable improvement in detecting long-period actions. When (c) both ICL and DAQ are applied to the baseline, there is a significant improvement in detecting actions with middle and long periods. However, it can be observed that while the MAE$_s$ for short actions improves, the OBO$_s$ decreases. This suggests that there is a decrease in the overall bias in counting predictions, but the accuracy of predicting exact counts diminishes. This could be attributed to the high number of short-period actions, which makes predicting exact counts particularly challenging.

## 4.5 QUALITATIVE RESULT

Figure 3 illustrates a representative example of the effectiveness of our proposed method on the RepCountA (Hu et al., 2022) dataset. The accuracy of our counting results is remarkably high, with precise alignment to the annotated action start and end positions. In contrast, the counting results obtained by the TransRAC (Hu et al., 2022) method are comparatively inferior, accompanied by a notable lack of interpretability in the predicted outcomes.

Figure 4 illustrates the generalization performance of the proposed method on the UCFRep (Zhang et al., 2020) dataset. The cross-dataset experiments demonstrate promising results. Notably, in the second example of Figure 4, where the action of soccer juggling is not present in the training set, our model still accurately recognizes the action instances, indicating robust generalization ability.

Please refer to our supplementary video for more qualitative results which are robust to varying cycle lengths and sudden interruptions.

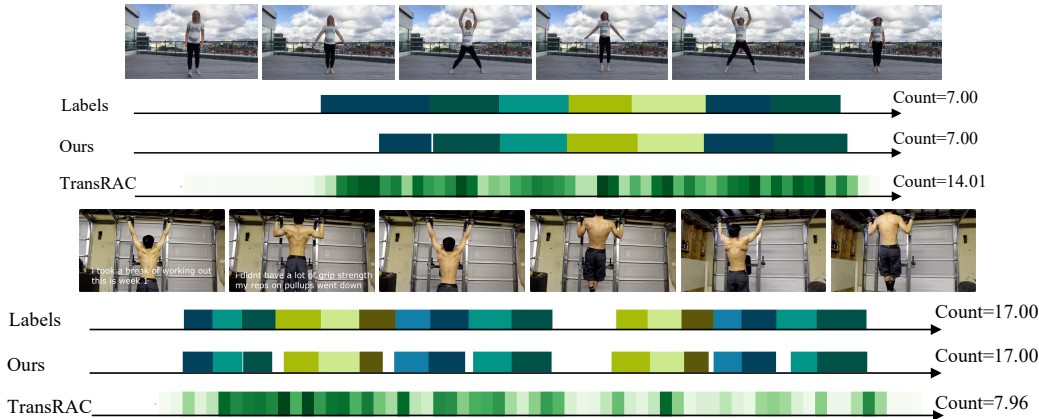

Figure 3: **RepCountA visualization results**. Each block represents a single GT or predicted action instance. TransRAC displays the results by density map, and the final count value is obtained by summing the values in the density map.

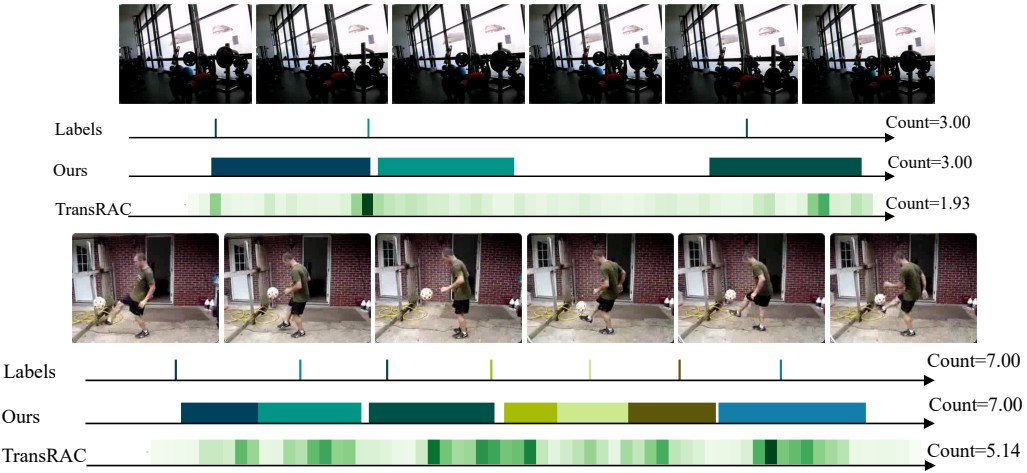

Figure 4: **UCFRep visualization results**. The vertical lines in the labels represent the time points at which the actions begin, for only the starting points of the actions annotated.

### 4.6 EFFICIENCY

To demonstrate the efficiency of our approach, we compare the model complexity between our method and the similarity matrix-based methods including RepNet (Dwibedi et al., 2020) and TransRAC (Hu et al., 2022), as detailed in Table 4. Thanks to the proposed query-based action cycle representation, our method can handle longer input video length while keeping a much smaller amount of parameters of the counting module.

## 5 CONCLUSION

In conclusion, we propose a novel approach for the TRC task that offers several advantages. Our method utilizes a simple yet effective representation for action cycles, reducing computational complexity and maintaining robustness across varying action periods and video lengths. The introduction of the DAQ improves generalization across different actions, while the ICL strategy facilitates learning repetitive action representations and distinguishing them from distractions. Experimental results on challenging benchmarks demonstrate the superiority of our approach, showcasing significant improvements compared to state-of-the-art methods. Besides, our method strikes an effective balance in handling diverse action periods and video lengths. As for **limitations,** due to the presence of only human motion videos in the dataset, our method currently focuses on human-centric videos. Also, there is still room for improvement in detecting short-period actions.

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

In this supplementary, we provide additional implementation details in Section 6 and more experimental results in Section 7. At last, we discuss the broader impacts and ethics in Section 8.

## 6    IMPLEMENTATION DETAILS

### 6.1    NETWORK ARCHITECTURE DETAILS

We adopt the classical Transformer (Vaswani et al., 2017) architecture for the encoder and decoder as detailed in Figure 5. To improve computational efficiency, we employ the deformable attention module (Zhu et al., 2020) in both the encoder and decoder. In the decoder, we employ the relation attention module (Shi et al., 2022) as the self-attention module and the deformable attention module (Zhu et al., 2020) as the cross-attention module. As for the reference points, we uniformly sample $T$ points in the timeline as reference midpoints in the encoder, while the reference points in the decoder are obtained from the midpoints of the action position in the previous layer. The initial midpoints are mapped from the learnable position embedding $\mathbf{Q}^{\text{pos}} \in \mathbb{R}^{Q \times C}$ to $\mathbb{R}^{Q \times 1}$ through an FC layer. The overall computational complexity is $O(kTC)$, where $k$ is the number of reference points in the deformable attention module.

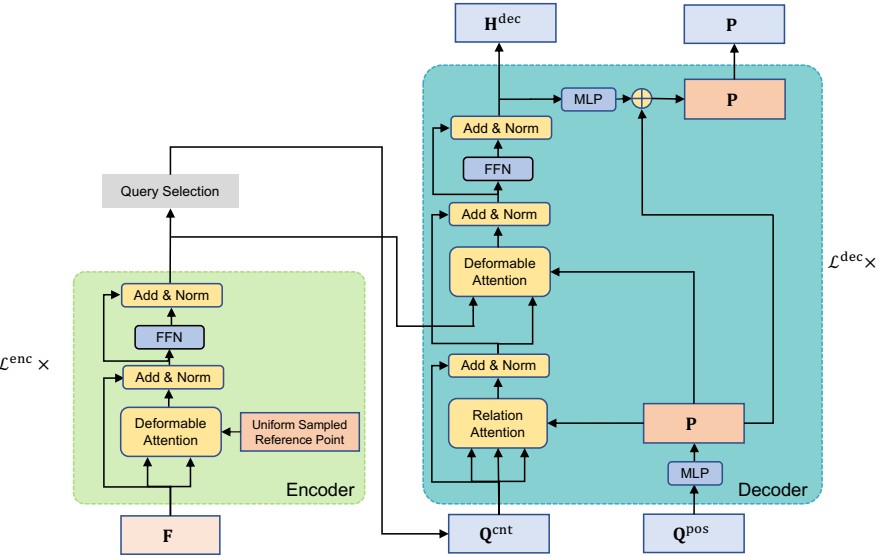

Figure 5: **Transformer architecture.**

**Query selection.**    We re-use the prediction heads to process the outputs of the encoder, with shared parameters. The top-$Q$ queries are those with the highest confidence score which are obtained from the classification head.

### 6.2    THE ROLE OF HUNGARIAN MATCHING

The Hungarian matching algorithm is employed for associating the $Q$ predicted instances from the decoder with the ground-truth (GT) results. Since the order of the $Q$ predicted instances from the decoder is not guaranteed, the Hungarian matching technique is used to establish a one-to-one correspondence between the labels and predictions. This correspondence is essential for computing the final loss. The matching score is a metric used to assess the relationship between the predicted instances and the GT.

We divide the matching score into two components, which evaluate whether the predicted positions and categories align with the GT. For the position component, we employ both Intersection over Union (IOU) and L1 distance to calculate scores that measure the proximity between predicted values and GT. Additionally, the Negative Log-Likelihood (NLL) loss is used to evaluate the accuracy of classification.

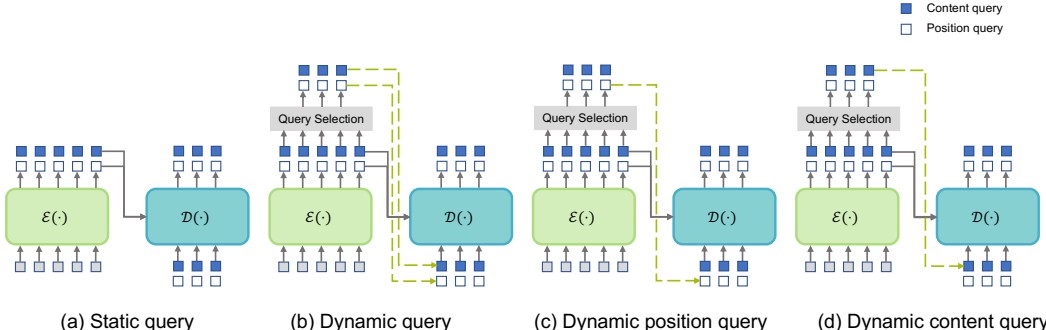

(a) Static query     (b) Dynamic query     (c) Dynamic position query     (d) Dynamic content query

Figure 6: **Different query type.** (a) Both the content query and position query are learnable variables that are optimized during the training process. (b) Both the content query and position query rely on the features of the encoder. (c) (d) Only the position query or content query relies on the encoder features, while the other one is a learnable variable.

Table 5: **Ablation results of different query type**. We illustrate each query type architecture in Figure 6.

|  | RepCountA | | UCFRep | |
|---|---|---|---|---|
|  | MAE ↓ | OBO ↑ | MAE ↓ | OBO ↑ |
| (a) Static query | 0.3542 | 0.4172 | 0.7447 | 0.1633 |
| (b) Dynamic query | 0.4042 | 0.3576 | 0.7142 | 0.1633 |
| (c) Dynamic position query | 0.5762 | 0.2649 | 0.6333 | 0.2653 |
| (d) Dynamic content query | **0.2809** | **0.4570** | **0.6016** | **0.2959** |

Ultimately, we construct a matching score matrix of dimensions $Q \times N_{gt}$, where $Q$ represents the number of predicted instances and $N_{gt}$ represents the number of GT instances. This matrix is then utilized in the Hungarian matching procedure.

## 6.3 TRAINING DETAILS

We implement the proposed model using PyTorch (Paszke et al., 2019) and train on a single NVIDIA V100 GPU. For training, we use an overlap rate of $0.75$ to segment the original videos. For testing, we use an overlap rate of $0.25$ and feed different segments of the same video as a batch to the network for prediction. Due to the overlap between segments, we add SoftNMS (Bodla et al., 2017) to alleviate the issue of multiple boxes pointing to the same action instance when testing.

## 7 ADDITIONAL EXPERIMENTS

### 7.1 ABLATION STUDY

**Different query type.** We explore several different query types illustrated in Figure 6. Table 5 shows the results of different query types on the RepCountA (Hu et al., 2022) dataset. The utilization of the position query (b) (c) yields suboptimal prediction results, which may be potentially attributed to inadequate initialization of position queries due to the substantial variations observed among action instances. Although the use of static query performs better on the RepCountA (Hu et al., 2022) dataset, it brings inferior results on the generalization test on UCFRep (Zhang et al., 2020) dataset. In contrast, our method (d) shows that by leveraging the information extracted from the encoder as content features, the decoder can concentrate on similar information, leading to notable improvements and enhanced generalization capabilities.

Table 6: **Ablation results of different updating strategies for the dynamic action query.** We additionally report MAE and OBO metrics for s̲hort-, m̲edium-, and l̲ong-period test sets of RepCountA (Hu et al., 2022) dataset.

| | MAE ↓ | OBO ↑ | MAE$_s$ ↓ | OBO$_s$ ↑ | MAE$_m$ ↓ | OBO$_m$ ↑ | MAE$_l$ ↓ | OBO$_l$ ↑ |
|---|---|---|---|---|---|---|---|---|
| (a) avg | 0.2987 | 0.4570 | 0.2378 | **0.2791** | 0.1953 | 0.4773 | 0.4107 | 0.5625 |
| (b) add | 0.3259 | **0.4636** | 0.2752 | 0.2093 | 0.2277 | **0.6136** | 0.4274 | 0.5312 |
| (c) linear | 0.3252 | 0.4503 | 0.2364 | **0.2326** | 0.2950 | 0.5455 | 0.4056 | 0.5312 |
| (d) direct | **0.2809** | 0.4570 | 0.2411 | 0.1628 | **0.1792** | 0.5455 | **0.3776** | **0.5938** |

Table 7: **Ablation results of different query number $Q$ on RepCountA (Hu et al., 2022) dataset.** We additionally report MAE and OBO metrics for s̲hort-, m̲edium-, and l̲ong-period test sets of RepCountA (Hu et al., 2022) dataset.

| | MAE ↓ | OBO ↑ | MAE$_s$ ↓ | OBO$_s$ ↑ | MAE$_m$ ↓ | OBO$_m$ ↑ | MAE$_l$ ↓ | OBO$_l$ ↑ |
|---|---|---|---|---|---|---|---|---|
| Q=10 | 0.6516 | 0.0795 | 0.5267 | 0.0233 | 0.5582 | 0.0682 | 0.7998 | 0.1250 |
| Q=20 | 0.2867 | 0.5364 | 0.2829 | 0.1860 | 0.1621 | 0.6591 | 0.3748 | 0.6875 |
| Q=30 | 0.2778 | 0.5033 | 0.2784 | 0.2093 | 0.1989 | 0.5909 | 0.3317 | 0.6406 |
| Q=40 | 0.2809 | 0.4570 | 0.2411 | 0.1628 | 0.1792 | 0.5455 | 0.3776 | 0.5938 |
| Q=50 | 0.3864 | 0.3974 | 0.2443 | 0.1860 | 0.1946 | 0.5000 | 0.6136 | 0.4688 |

**Updating strategy of dynamic action query.** We compare different ways of how to update the action query in the decoder on the RepCountA (Hu et al., 2022) dataset in Table 6. Our method (d) differs from the other three baselines in how to update the content query.

Baseline (a) takes the average of the features produced by the query selection module and assigns it to the content query for the decoder. Baseline (b) takes the sum of the encoded feature and a learnable embedding to update the content query for the decoder. Baseline (c) first concatenates the encoded features with a learnable embedding and then transforms it with a linear matrix to update the content query. Our method (d) directly uses the encoded features after the query selection as the content query for the decoder. The four implementations are as follows:

$$\text{(a)} \quad \mathbf{Q}^{\text{cnt}} = \frac{1}{Q} \sum_Q \mathbf{H}^{\text{sel}},$$

$$\text{(b)} \quad \mathbf{Q}^{\text{cnt}} = \mathbf{H}^{\text{sel}} + \mathbf{Q}^{\text{l}},$$

$$\text{(c)} \quad \mathbf{Q}^{\text{cnt}} = \mathbf{W}\,\text{Concat}(\mathbf{H}^{\text{sel}}, \mathbf{Q}^{\text{l}}),$$

$$\text{(d)} \quad \mathbf{Q}^{\text{cnt}} = \mathbf{H}^{\text{sel}},$$

where $\mathbf{W} \in \mathbb{R}^{C \times 2C}$ denotes a learnable weight matrix, $\mathbf{H}^{\text{sel}} \in \mathbb{R}^{Q \times C}$ is the feature produced by the query selection module, and $\mathbf{Q}^{\text{l}} \in \mathbb{R}^{Q \times C}$ is a learnable embedding.

As shown in Table 6 (d), direct updating performs the best on middle- and long-period actions. Because this method focuses on the most salient features, features associated with long-period actions are more prominent in time compared to short-period actions, making them easier to recognize. (b) addition and (c) linear combination have worse MAE. Because of the presence of learnable embeddings, predictions may exhibit a certain bias during inference, resulting in the prediction of values that deviate significantly from the true values.

**Query number $Q$.** We conduct ablation experiments on different choices of the query number $Q$. The experimental results on the RepCountA (Hu et al., 2022) and UCFRep (Zhang et al., 2020) datasets are presented in Table 7 and 8, respectively.

It can be observed that a query number of $Q = 40$ strikes a good balance between the two datasets. When the query number is too small, such as $Q = 10$, the results obtained are expectedly inferior. However, as the number of queries increases, the performance improves on the RepCountA (Hu et al., 2022) dataset while degrading on the UCFRep (Zhang et al., 2020) dataset. This discrepancy can be

Table 8: **Ablation results of different query number $Q$ on UCFRep (Zhang et al., 2020) dataset.** We additionally report MAE and OBO metrics for short-, medium-, and long-period test sets of UCFRep (Zhang et al., 2020) dataset.

|       | MAE ↓  | OBO ↑  | $MAE_s$ ↓ | $OBO_s$ ↑ | $MAE_m$ ↓ | $OBO_m$ ↑ | $MAE_l$ ↓ | $OBO_l$ ↑ |
|-------|--------|--------|-----------|-----------|-----------|-----------|-----------|-----------|
| Q=10  | 0.9543 | 0.0204 | 0.9505    | 0.0000    | 0.9437    | 0.0500    | 0.9882    | 0.0000    |
| Q=20  | 0.7933 | 0.1429 | 0.8483    | 0.0244    | 0.8057    | 0.1250    | 0.6314    | 0.4706    |
| Q=30  | 0.7416 | 0.2143 | 0.8505    | 0.0488    | 0.6789    | 0.2750    | 0.6265    | 0.4706    |
| Q=40  | 0.6016 | 0.2959 | 0.7069    | 0.0488    | 0.5777    | 0.4250    | 0.4039    | 0.5882    |
| Q=50  | 0.5621 | 0.3571 | 0.7369    | 0.0244    | 0.4651    | 0.5250    | 0.3686    | 0.7647    |

Table 9: **Ablations on intermediate supervision on the RepCountA (Hu et al., 2022) dataset.** We additionally report MAE and OBO metrics for short-, medium-, and long-period test sets.

|                | MAE ↓      | OBO ↑      | $MAE_s$ ↓  | $OBO_s$ ↑  | $MAE_m$ ↓  | $OBO_m$ ↑  | $MAE_l$ ↓  | $OBO_l$ ↑  |
|----------------|------------|------------|------------|------------|------------|------------|------------|------------|
| Baseline       | **0.2809** | **0.4570** | **0.2411** | **0.1628** | **0.1792** | **0.5455** | **0.3776** | **0.5938** |
| - inter. superv. | 0.5062   | 0.2583     | 0.4792     | 0.0930     | 0.3554     | 0.2500     | 0.6281     | 0.3750     |

attributed to the different counting distributions present in these two datasets. Therefore, based on our experiments, we choose $Q = 40$ as the optimal value.

**Intermediate supervision.** To accelerate training and promote performance, we apply the total losses for both the final encoder layer and each decoder layer. We conduct the ablation experiments where we only keep the total losses at the final decoder layer and remove all the intermediate supervision. The results are shown in Table 9, where removing the intermediate supervision degrades the performance.

**Confidence threshold $\alpha$.** We empirically set the action instance confidence threshold $\alpha$ as 0.2. We assess the impact of different action confidence thresholds on the final counting results in Figure 7 (orange curve). We can see that setting the threshold between 0.2 to 0.4 yields similar performance for our method and surpasses TransRAC (Hu et al., 2022) by a large margin.

**Feature dimension $C$.** We conduct ablation experiments on the feature dimension $C$ on the RepCountA (Hu et al., 2022) dataset, and the results are presented in Table 10. It can be observed that the best results are obtained when the dimension is set to $C = 512$. When $C$ is smaller, the model struggles to handle long-period instances effectively, while larger values of $C$ lead to inaccurate predictions for short-period instances. When $C$ is small, it becomes challenging for the model to accommodate longer sequences of information within the limited capacity. On the other hand, when $C$ is large, the model becomes overloaded with excessive information, leading to redundant information for short-period actions and hindering accurate prediction results.

## 7.2 QUALITATIVE RESULTS

We show more qualitative results in Figure 8, where our method can handle various action types robustly. We compare our results with the state-of-the-art method TransRAC (Hu et al., 2022), which performs inferior. One potential reason could be that the predicted density map of TransRAC (Hu et al., 2022) lacks interpretability, making the wrong count. Please refer to the `video-3031.mp4` for more video results that are robust to varying cycle lengths and sudden interruptions.

Figure 9 shows two typical failure cases. In the first case, due to a change in viewing perspective, the person is truncated, making a large difference in the action feature, resulting in several missed cycle counts. In the second case, when the action speeds up later in the video, our method mistakenly identifies one action cycle as two, resulting in an overcount.

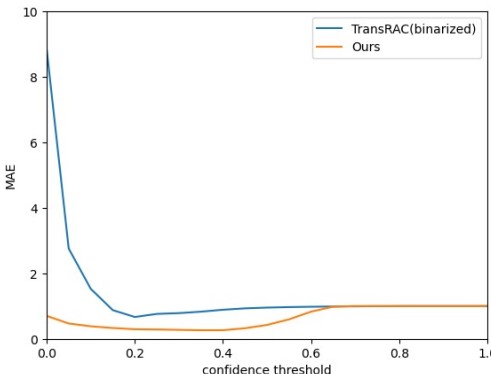 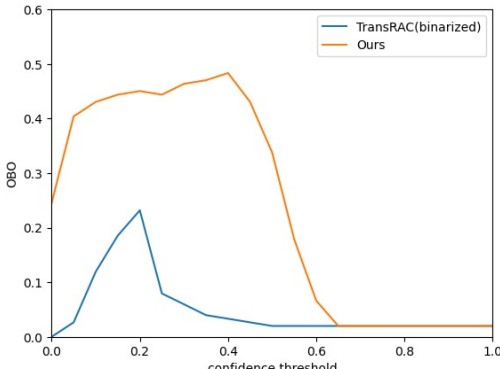

Figure 7: **Ablations of different confidence thresholds of our method and TransRAC (Hu et al., 2022).** We depict the MAE (left) and OBO (right) curves of our method (orange curve) and the TransRAC (blue curve) approach with regard to different confidence thresholds. The metrics of TransRAC are obtained by binarizing the density map of TransRAC output and then summing to obtain the final count.

Table 10: **Ablation results of different feature dimensions on the RepCountA (Hu et al., 2022) dataset.** We additionally report MAE and OBO metrics for short-, medium-, and long-period test sets of RepCountA (Hu et al., 2022) dataset.

| $C$ | MAE ↓ | OBO ↑ | $MAE_s$ ↓ | $OBO_s$ ↑ | $MAE_m$ ↓ | $OBO_m$ ↑ | $MAE_l$ ↓ | $OBO_l$ ↑ |
|---|---|---|---|---|---|---|---|---|
| 128 | 0.4514 | 0.3709 | 0.2361 | 0.2093 | 0.2750 | 0.4545 | 0.7175 | 0.4219 |
| 256 | 0.3298 | **0.4570** | **0.2207** | **0.2326** | 0.2366 | **0.6136** | 0.4673 | 0.5000 |
| 512 | **0.2809** | **0.4570** | 0.2411 | 0.1628 | **0.1792** | 0.5455 | **0.3776** | **0.5938** |
| 1024 | 0.6259 | 0.2848 | 0.6241 | 0.0465 | 0.3562 | 0.2955 | 0.8126 | 0.4375 |

## 7.3 LIMITATIONS

We observe that the proposed method performs well in predicting medium- and long-period actions, but its performance is still not satisfactory for short-period actions. This phenomenon is similar to the situation in object detection where detecting small objects is more difficult than detecting large ones (Carion et al., 2020), where the core reason may lie in the ability of the neural network to extract salient features of the target. Compared to large objects, CNN-based neural networks do not pay enough attention to small objects. Therefore, as in traditional object recognition, feature pyramid methods can be used to enhance the features of small objects and improve their recognition. We may also introduce this multi-scale strategy to address the issue in the future.

## 7.4 VISUALIZATION OF DAQ

To investigate the role of DAQ in practical scenarios, we selected three test videos from RepCountA and visualized the features output by the encoder using the t-SNE method in Figure 10. In this visualization, the features selected by the query selection module and those not selected are represented by orange and blue points, respectively. It can be observed that features belonging to the background exhibit a clustered pattern. Features corresponding to actions are dispersed around, while the features selected by DAQ are situated in the middle part of the action-related features. By using this subset of features as the initial values for $\mathbf{Q}^{cnt}$, the model can better focus on prominent actions in the video, thereby enhancing the performance and generalization of the model.

## 8 BOARDER IMPACTS AND ETHICS

We use public datasets in our experiments following their licensing requirements. This work has no harm to society for not explicitly using biometrics and outputting a number that only represents the

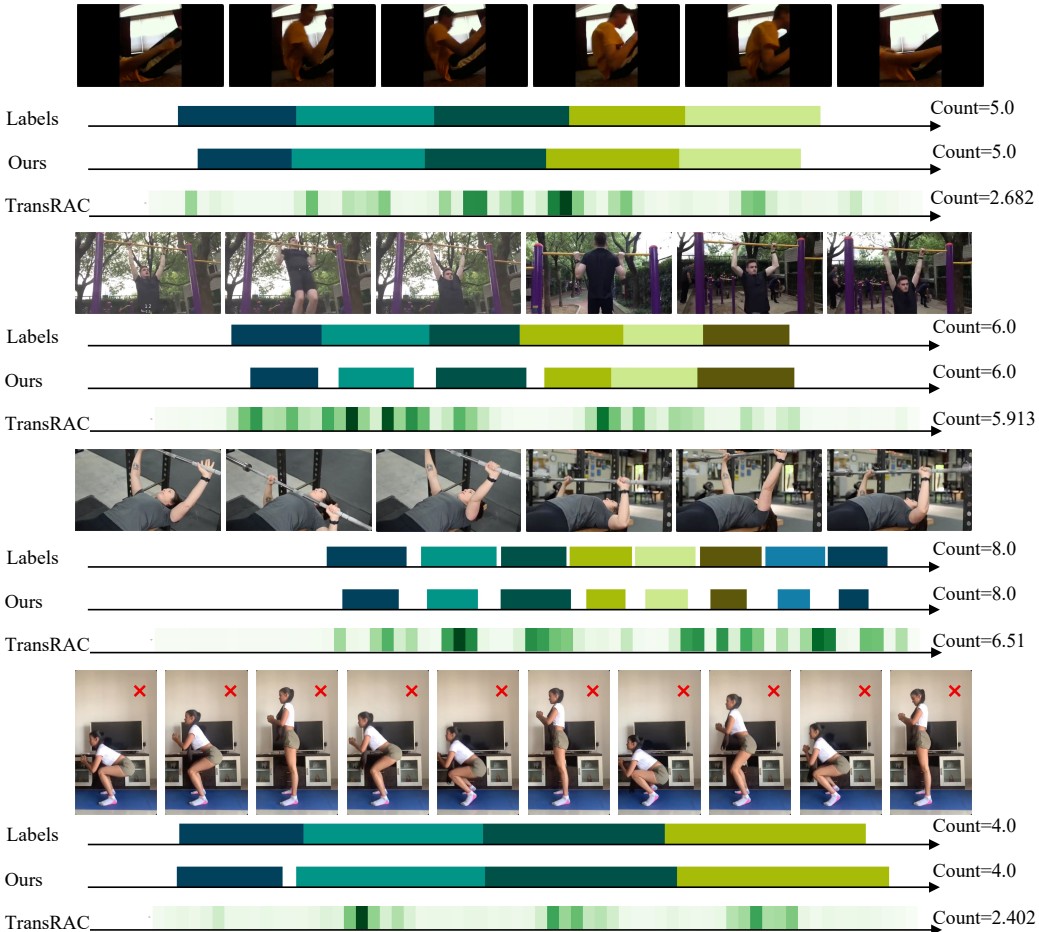

Figure 8: **Visualization of qualitative results on RepCountA (Hu et al., 2022) dataset.** Each block represents a single annotated or predicted action instance. TransRAC (Hu et al., 2022) displays the results of its density map, and the final count value is obtained by summing the values in the density map.

repetition cycles. The method proposed in this paper does not violate ethical principles and strictly adheres to standards and regulations.

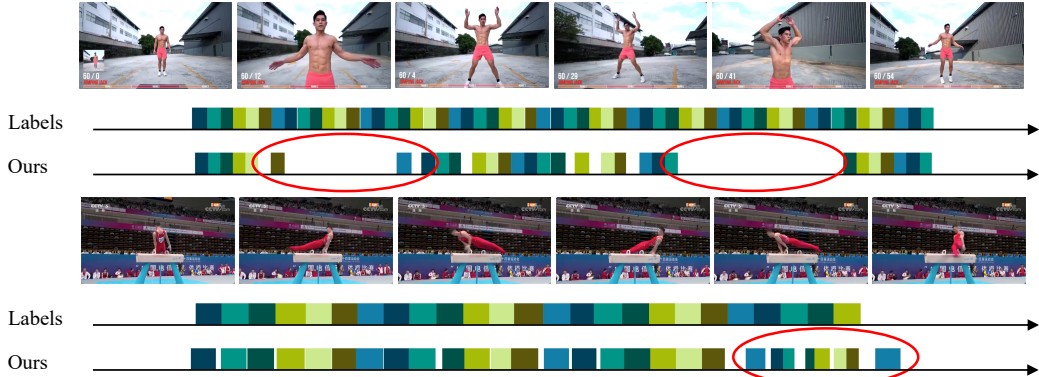

Figure 9: **Visualization of failure case on RepCountA (Hu et al., 2022) dataset.** In the first case, body truncation caused by the camera zooming in leads to miscounting. In the second case, the high speed of movement leads to overcounting.

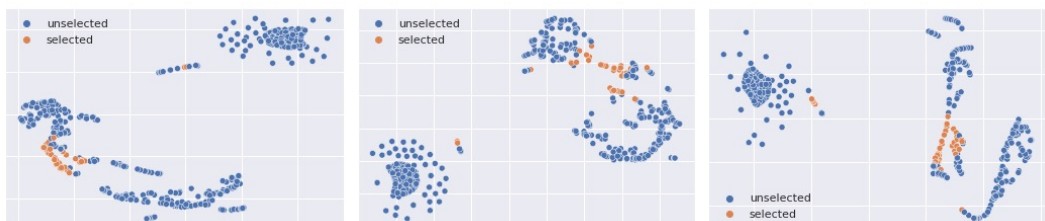

Figure 10: **Visualization of dynamic action query of three cases.** We conduct t-SNE analysis on the content query $\mathbf{Q}^{\text{cnt}} \in Q \times C$ obtained from the encoder of three samples. In the resulting 2-dimensional space, the selected and unselected queries are represented by orange and blue dots, respectively. Our dynamic action queries exhibit a clear separation into two distinct clusters (actions vs. null). And the three different cases possess different patterns, validating the effectiveness of the dynamic query updating design.

