# OpenReview forum: "Temporal Repetition Counting with Dynamic Action Queries"
_ICLR.cc/2024/Conference — ICLR 2024 Conference Withdrawn Submission_

### Official Review · Reviewer_rXK7 · 2023-10-28

**Soundness:** 3 good
**Presentation:** 3 good
**Contribution:** 2 fair
**Rating:** 3
**Confidence:** 4

**Summary:**

In this work, the authors mainly study the temporal repetition counting task and propose a new action query representation to replace the directly used features. The dynamic action query is used to guide the model to focus on the action of interest targeting for the open-set setting. The inter-query contrastive learning is designed to learn repetitive action representation. The experimental results show the effectiveness of their proposed model.

**Strengths:**

This paper is well-written and easy to follow. The idea of using action query to replace directly calculating similarity matrix based on features makes sense. This way could greatly reduce the computational complexity and the authors also show the comparisons to validate it. They have done comparison and ablation studies to validate their proposed components.

**Weaknesses:**

The main contributions of this work include (1) the new representation, (2) dynamic action query, and (3) inter-query contrastive learning, which might be merged into one or two. The related paragraph in Section 3, should be added more details, for example, Sec 3.3 and 3.4. Besides, for the claimed contribution inter-query contrastive learning might not be a sufficient contribution, just using contrastive learning.

For Table 4, why not the authors use the same setting for comparison, adding TSN under 64, or RepNet and TransRAC under 513 length? Why separate the backbone and counting part, also why does the backbone use per frame and the counting part use all frames?
For the visualization, the authors might try to show some examples of the content query and position query. It might not be clear or need some support for two kinds of queries, content and position.

**Questions:**

See weakness.

---

### Official Review · Reviewer_t7e9 · 2023-10-30

**Soundness:** 3 good
**Presentation:** 3 good
**Contribution:** 2 fair
**Rating:** 3
**Confidence:** 5

**Summary:**

This paper tackles the problem of temporal repetition counting, which aims to identify the number of repetitve actions in videos. The proposed method follows the DETR-alike query-based detection pipeline and proposes two improvements of dynamic query initialization and inter-query constrastive learning to facilitate the open-set action counting. Experiments on popular benchmark shows large improvement over previous literature and demonstrates the effectiveness of the proposed methods.

**Strengths:**

+ The paper is generally well-writen and easy to follow.
+ Experiments show superior performance of the proposed method on two popular TRC benchmarks.
+ The evaluation for short, medium and long action durations are good metrics to show the performance on different action periods for all methods.
+ Ablation study supports the effectiveness of its proposed modules.

**Weaknesses:**

- The novelty of the model design appears to be limited.

  - Generally, the proposed method follows the well-studied DETR-alike architecture to solve a temporal detection task, so at architecture-level, the method looks conventional.
  - The idea of decomposing queries into content and position vectors has been studied in DAB-DETR [M1], DINO[M2], etc. The dynamic query initialization is also explored in object detection literature. These designs look very familiar to me.

- Unfair comparison in experiments.

  - It is surprising to see the baseline model in tab.3 already outperforms all models in tab. 1, including the state-of-the-art TAD methods. As the proposed baseline design is no complex than TadTR, it appears that the authors' adaptation may have led to a performance degradation, possibly due to the domain gap between class-dependent TAD and class-agnostic TRC. The authors should consider competing against stronger models or compare with class-agnostic temporal detection / segmentation methods, such as RTD-Net [M3], BMN [M4].
  - The backbone network used here is TSN and VideoMAE, while the compared methods in tab.1 use backbones including Resnet50, Video Swin, I3D, etc. As the backbone network is important to downstream detection performance, the current comparison is quite messy and unfair regarding the backbone choices. The authors are suggested to align backbone network with the start-of-art-methods and at least add a column in comparison tables to specify the backbone network used for all methods.

- Factual Mistake and Missing citation in Introduction: The mention of Actionformer employing query-based detection in TAD is incorrect. In addition, there misses the mention of other important query-based TAD framework ([M3]).

  [M1] Liu, Shilong, et al. "DAB-DETR: Dynamic Anchor Boxes are Better Queries for DETR." *International Conference on Learning Representations*. 2021.

  [M2] Zhang, Hao, et al. "DINO: DETR with Improved DeNoising Anchor Boxes for End-to-End Object Detection." *The Eleventh International Conference on Learning Representations*. 2022.

  [M3] Tan, Jing, et al. "Relaxed transformer decoders for direct action proposal generation." *Proceedings of the IEEE/CVF international conference on computer vision*. 2021.

  [M4] Lin, Tianwei, et al. "Bmn: Boundary-matching network for temporal action proposal generation." Proceedings of the IEEE/CVF international conference on computer vision. 2019.

**Questions:**

- What's the backbone network used for the re-implementation of Actionformer and TadTR in TRC task?
- Why the input length in efficiency table (tab.4) is different for RepNet, TransRAC and the proposed method, since all of these methods use sliding window for preprocess and can adjust input length in theory?
- What's the possible reason that the proposed method achieves weaker results for short actions ion RepCountA dataset?

---

### Official Review · Reviewer_Xbom · 2023-11-01

**Soundness:** 3 good
**Presentation:** 4 excellent
**Contribution:** 2 fair
**Rating:** 5
**Confidence:** 4

**Summary:**

This paper proposes an approach for Temporal Repetition Counting (TRC), which claimed to be linear complexity as opposed to existing methods by avoiding to compute covariance-matrix explicitly. The proposed approach provides two novel components: Dynamic Action Query (DAQ) and Inter-query Contrastive Learning (ICL). Experiments are conducted on RepCountA (for training and testing) and UCFRep (for testing only). In terms of novelty, the paper has limited novelty, as the paper mainly contribute a new architecture for TRC (not proposing the problem nor introducing dataset). In terms of experiments, the experimental results are also not strong enough. This mainly because of (i) small datasets were used which may raise concerns about the applicability of the proposed approach on larger datasets or practical applications; (ii) ablation experiments are moderate, and do not convey any interesting finding beyond some improved metrics on small benchmarks. In terms of written presentation, the paper is well-written and easy to read and understand.

**Strengths:**

- The paper has been very well-written and organized making it easy to read and understand.
- The proposed approach is technically sounded.
- The paper provides a good set of diverse baselines, also re-implement some baseline from temporal action detection domain.

**Weaknesses:**

- The technical novelty brought by the paper is limited or moderately low. More concretely, the paper contributes a new architecture for TRC (Fig 2) which two main features: DAQ & ICL. The main claim from the paper is about complexity which is linear instead of quadratic as a result of not need a similarity matrix as input. Note that the paper does not propose a new problem or a new dataset as this is well-studied problem. Also note that this paper is not the first one to apply transformer to TRC (RepNet & TransRAC have already done that). The benefit may be only on being a similarity-matrix free approach.
- The problem is small and very specific. Although the reviewer admits that the problem of TRC is interesting, the scope of impact of TRC is much smaller than video classification or action detection making the impact of this work smaller.
- Experiments are done on small datasets. On RepCountA, models are trained on 757 videos and tested on 130 videos. On UCFRep, models are tested on 106 videos. This raises a question if the proposed approach will work on larger dataset or real practical applications.
- The main claim of the proposed approach is about reduced complexity, However it presented very limited experiments to show case this benefit. There is only one table 4 conveying this benefit. However, the table only provide number of parameters and theory FLOPs and not reporting real runtime. In addition, the results seems missing, as the paper presents only Ours with TSN, but the Ours with VideoMAE, which is supposed to have better performance.

**Questions:**

- As mentioned above, my justification is multiple aspects, in general, it is OK to have paper with low novelty but strong empirical results with new / useful insights OR significant novelty with moderate experimental results. Unfortunately, this paper did not make any of these aspects significantly enough to meet the top-tier conference bar. The reviewer is open to hear the rebuttal from the author(s).

* Some other comments
- In section 3.4, the paper attributes the Hungarian matching algorithm to Carion et al. 2020 which is not correct. It should be way earlier than that, it should be either cite to the correct textbook or rephrase the text, e.g., Following DETR, we apply ...
- In table 1 & 2, are the baselines using the same backbone features as the proposed approach?
- In section 4.3 (bottom line), the text mentions to have 70.8% improvement on OBO, but the reviewer could not find the same gap in Table 1 or 2, is it a typo?
- Table 4, please add also Ours (VideoMAE).